# Can Hyperbaric Oxygen Therapy Be Used to Treat Children after COVID-19? A Bibliographic Review

**DOI:** 10.3390/ijerph192215213

**Published:** 2022-11-18

**Authors:** Andrzej P. Myśliwiec, Julia T. Walatek, Anna Tarnawa, Katarzyna Nierwińska, Iwona Doroniewicz

**Affiliations:** 1Laboratory of Physiotherapy and Physioprevention, Institute of Physiotherapy and Health Sciences, Academy of Physical Education in Katowice, 40-065 Katowice, Poland; 2Physiotherapy Center “Galen Rehabilitation”, 43-150 Bieruń, Poland; 3Center for Intensive Rehabilitation of Children “Michałkowo”, 43-360 Wilkowice, Poland

**Keywords:** hyperbaric oxygen therapy (HBOT), COVID-19, children

## Abstract

The coronavirus disease (COVID-19) epidemic is a public health emergency of international concern. It was believed that SARS-CoV-2 virus was much less likely affect children. Statistics show that children account for 2–13% of all COVID-19 patients in individual countries. In the youngest population, acute respiratory failure is not as serious a problem as complications after COVID-19, mainly pediatric inflammatory multisystem syndrome (PIMS, MIS-C). This study used a bibliography review. The Medline database (using the PubMed platform) and the Cochrane Clinical Trials database were searched using the following keywords: hyperbaric oxygen therapy for children, treatment of children with COVID-19, and use of HBOT in the treatment of children following COVID-19. Thirteen publications that quantitatively and qualitatively described the efficacy of HBOT application in the treatment of pediatric diseases were eligible among the studies; those relating to the use of HBOT in the treatment of children with COVID-19 and its complications were not found. The bibliographic review showed that hyperbaric oxygen therapy can be used in the treatment of children after carbon monoxide poisoning, with soft tissue necrosis, bone necrosis, after burns, or after skin transplant. No evidence supported by research has been found in scientific journals on the effectiveness of the use of hyperbaric oxygen therapy in children with a history of COVID-19 infection. Research data are needed to develop evidence-driven strategies with regard to the use of HBOT therapy in the treatment of children and to reduce the number of pediatric patients suffering because of complications after COVID-19.

## 1. Introduction

The global COVID-19 pandemic has led to a significant increase in the number of patients suffering from respiratory conditions, cognitive impairment, and chronic fatigue, which are direct consequences of COVID-19 [1,2,3]. Initially, it was believed that the SARS-CoV-2 virus was much less likely to affect children, but studies have repeatedly shown that they are just as susceptible as adults, although the course of COVID-19 is mostly asymptomatic in them [4,5,6]. Statistics show that children account for about 2–13% of all COVID-19 patients in individual countries, but this figure may be a significant underestimate due to the small number of tests among the youngest [4,5,6,7,8]. However, t in the most severe cases, children also develop acute respiratory failure, which is characterized by a reduced ratio of the partial pressure of oxygen to the fraction of inspired oxygen and reduced blood oxygen saturation [9]. However, a much bigger problem in the youngest population is complications after COVID-19, mainly involving pediatric inflammatory multisystem syndrome (PIMS, MIS-C) [9,10,11]. The characteristic feature of PIMS is generalized multisystem inflammation with co-morbid fever, skin and mucosal lesions, and cardiopulmonary and gastrointestinal symptoms appearing several weeks after infection [9,11]. PIMS can also lead to the development of complications such as shock, coronary artery aneurysms, and acute myocarditis, which is often confirmed by echocardiographic abnormalities [12]. To date, no method has been developed to unequivocally show improvements in the vital signs of COVID-19 patients. Sometimes supportive measures such as oxygen therapy with intranasal cannulas or masks, mechanical and non-invasive ventilation or extracorporeal blood oxygenation do not improve patient oxygenation [13].

In this situation, an effective way to increase blood oxygen saturation could be the use of hyperbaric oxygen therapy (HBOT), which not only raises the partial pressure of oxygen in the blood but also prevents the activation of inflammatory cells [14], which would probably work well in the treatment of pediatric inflammatory multisystem syndrome, which is a direct complication of COVID-19 in the youngest [11]. HBOT is a treatment method that involves treating the patient with 100% oxygen with increased pressure in a specially constructed hyperbaric chamber. The pressure exerted during the exposure is expressed as the sum of the atmospheric pressure and the pressure prevailing in the chamber. The pressure exerted during the exposure is expressed as the sum of the atmospheric pressure and the pressure prevailing in the chamber. It is considered that it should be at least 1.4 absolute pressure (ATA). The HBOT intervention contributes to a significant increase in blood oxygen pressure by oxygenating hemoglobin and dissolving it in blood plasma. These processes significantly increase the extent of oxygen diffusion from the capillaries to the surrounding hypoxic cells, thus improving their metabolism [15]. Hyperbaric chambers also find application in many areas of pediatrics and numerous pediatric rehabilitation centers, which would economically enable the introduction of the new COVID-19 adjunctive treatment method without the need to purchase additional equipment. The aim of the present paper was to review the literature in search of clinical studies evaluating the extent of use and the validity and effectiveness of hyperbaric oxygen therapy (HBOT) in the treatment of pediatric patients and to indicate the validity of HBOT as a possible method of treatment and recovery for children after COVID-19 and its immediate complications.

## 2. Materials and Methods

The article presents a bibliographic review based on the following steps: formulation of research questions, development of a literature search strategy, the search of bibliographic databases, and selection of studies for the review based on pre-established inclusion and exclusion criteria from the review, followed by data extraction, analysis, and interpretation. Due to the high heterogeneity of the studies included in the review, qualitative assessment of publications and data synthesis by meta-analysis were omitted. A question was asked about the possibility of using HBOT in the treatment of children, in particular after a history of COVID-19 infection. The Medline database (using the PubMed platform) and the Cochrane Clinical Trials database were searched using the following keywords: hyperbaric oxygen therapy for children, treatment of children with COVID-19, and use of HBOT in the treatment of children following COVID-19.

### Inclusion/Exclusion Criteria

Clinical and observational studies on the use of HBOT in the treatment of pediatric patients were considered inclusion criteria. The selected studies used any HBOT protocols and methods to assess health effects. Papers published in Polish and English were reviewed. Reports, case studies, and studies involving adults were excluded. Other exclusion criteria were studies on children with disease entities for which the ECHM issued a negative recommendation, i.e., “a strong recommendation not to use HBOT” [16].

## 3. Results

Thirteen publications that quantitatively and qualitatively described the efficacy of HBOT application in the treatment of pediatric diseases were eligible for review (Figure 1). Among the studies, those relating to the use of HBOT in the treatment of children with COVID-19 and its complications were not found.

### 3.1. Study Characteristics

A detailed summary of the studies discussing carbon monoxide poisoning, necrotizing soft-tissue infections, osteonecrosis, thermal burns, and skin grafts is provided in Table 1.

### 3.2. Results of Individual Studies

#### Carbon Monoxide Poisoning

One study retrospectively evaluated 74 children (aged 1.0–17.8 years) who were exposed to CO. They were hospitalized at Eskişehir Osmangazi University in the pediatric ward from 1 June 2003 to 1 June 2005. In all patients, blood tests and additional examinations were performed immediately after admission. Blood analysis was performed with a critical parameter analyzer (radiometer). Since all patients were non-smokers, elevated carboxyhemoglobin (COHb) levels were defined as above 2%. All patients received normobaric oxygen therapy (NBOT). Oxygen was administered through a mask at a rate of 10 L per minute until the COHb level fell below 2%. In addition, 38 of the 74 patients also underwent HBO therapy, for which the indications were present neurological symptoms during admission (seizures, coma, fainting) or further neurological changes after NBO therapy (headache, visual disturbances, ataxia) and COHb levels above 20%. Hyperbaric oxygen therapy was started within 24 h after exposure to CO in a multiplace chamber (Bara-Med^®^, Model HTC 4/2/6, Environmental Tectonics Corporation, Southampton, PA, USA). Each session in the chamber lasted about 140 min, including 20 min of compression, 100 min at 2.4 ATA with two 5-min breaks every 30 min, and 20 min of decompression. In patients who received HBO therapy, the time of COHb return to less than 2% ranged from 4 to 52 h. Initial COHb levels were significantly higher in patients with abnormal neurological symptoms. Fifty-seven patients were followed up in consecutive months (6.1–12.5 months). One of them had visual disturbances after being discharged from the hospital and developed symptoms of epilepsy seven months later. The disturbances resolved within a year, but some abnormalities were detected during MRI. No complications were detected in the remaining patients. Researchers suggest that the use of HBO therapy is safe for children. It can also be effective against delayed neurological complications [24]. Another study retrospectively evaluated 150 children who were treated after carbon monoxide poisoning at Jacobi Medical Center in New York between August 1992 and March 1996. All children had COHb above 25% or below 25% at the same time, showing significant neurological, cardiac, or respiratory abnormalities. All of the patients included in the study were under the age of 18, with an average age of 7.2 years. In this group, 59.9% of patients had carbon monoxide poisoning, while the remaining 40.1% had both carbon monoxide poisoning and suffered from smoke inhalation due to fire. The study did not specify the HBOT parameters that were used. The study showed that children with high COHb levels but no additional risk factors who received HBOT had no risk of death. In conclusion, preliminary data suggest that children with only carbon monoxide poisoning who were treated with HBOT had a low risk of death, regardless of initial COHb levels. Furthermore, children with carbon monoxide poisoning and smoke poisoning were significantly more likely to die than those with carbon monoxide poisoning alone. Associated risk factors for death in patients with smoke inhalation and carbon monoxide poisoning include low temperature upon arrival in the ED, high carboxyhemoglobin levels, and cardiopulmonary arrest [27]. Another study was conducted prospectively between March 2015 and April 2016 in patients admitted to the Pediatric Emergency Department. People under the age of 19 were eligible for the study. In the study, carbon monoxide poisoning was diagnosed by interview, clinical results, and COHb levels. All patients received NBO treatment after admission. On admission to the ward, they had neurological symptoms (loss of consciousness, altered mental status, seizures), cardiac abnormalities, COHb levels higher than 25%, and elevated troponin T levels. Patients treated with HBO had blood and urine samples collected on admission (T1), at the sixth hour (T2), before HBO therapy (T3), and after HBO therapy (T4). Samples were taken as soon as possible, at most one hour after HBO therapy. If the patient was not treated with HBO in the first 6 h, sampling at the sixth hour (T2) was considered a sample before HBO therapy (T3). We compared the initial (T1) level of oxidative stress and the level of antioxidant parameters in all patients with CO poisoning. During the study period, 54 children were diagnosed with carbon monoxide poisoning. However, five patients whose samples were not collected in a timely manner and two patients who were admitted to the emergency department after HBO treatment were excluded from the study. Sixteen patients were treated with HBO. None of the patients were treated with HBO within the first 6 h of poisoning. This study showed that CO poisoning is associated with increased lipid peroxidation in children immediately after poisoning. However, neither NBOT nor HBOT has a significant effect on oxidative stress or levels of antioxidant parameters, except for catalase activity. They noted the need to do further research into the possibility of treating carbon monoxide poisoning with HBOT and NBOT [22]. Another retrospective study conducted from January 2004 to March 2014 included patients of Hacettepe İhsan Doğramacı Pediatric Hospital aged 0 to 18 years diagnosed with CO poisoning. Patients’ demographic characteristics, characterization of symptoms, GCS score, laboratory results, treatment, clinical course, and results in the acute period were recorded. Neurological abnormalities were defined as altered consciousness, seizures, or abnormal neurological examination results on admission. Cardiac abnormalities were defined as tachycardia, hypotension, perfusion abnormalities, or a weak pulse on admission. Elevated troponin T levels above 0.014 mg/mL and failure of 2 or more organs were defined as multi-organ failure. All patients were treated with NBOT. Normobaric oxygen is 100% oxygen administered through a mask at a pressure of 1 ATA. Hyperbaric oxygen was administered to patients with neurological symptoms on admission, such as loss of consciousness, collapse, seizures, and COHb levels higher than 25%, or cardiac abnormalities such as hypotension, increased troponin T levels, ischemic ECG changes, or abnormal echocardiography results. Hyperbaric oxygen was administered at 5 ATA for 90 min. Of all patients, 28% (*n* = 93) were treated with HBOT and 72% (*n* = 238) with NBOT. No side effects of HBO therapy were observed. Hyperbaric oxygen facilitates the elimination of CO from the body and increases the partial pressure of oxygen in the arteries and tissues. It also modulates the inflammation of processes caused by CO poisoning. The study concluded that hyperbaric oxygen therapy should be used in all cases of acute symptomatic CO poisoning [17].

### 3.3. Necrotizing Soft-Tissue Infections

Another study looked at HBO therapy in the treatment of patients with necrotizing soft-tissue infections (NSTI). The benefits of using hyperbaric oxygen therapy to treat deep soft-tissue infections, including the treatment of children, were confirmed at the Burn Treatment Center in Siemianowice Ślaskie from 2002 to 2006. Forty-three patients between the ages of 10 months and 86 years were treated in the chamber, including 18 patients with soft tissue infections (hard-to-heal wounds), 14 patients with post-traumatic soft tissue necrosis, two patients with crush syndrome in the hands, lower limbs, and feet, seven patients with gas gangrene, one patient with anaerobe infection after nephrectomy, and one patient with soft-tissue necrosis of the amputation stump. An analysis of the clinical material based on the diagnoses was performed, and the results obtained were analyzed and compared with a group of patients treated during the same period and for the same reason but without HBO. In a group of patients with deep soft tissue infection, post-traumatic crush syndrome, and gas gangrene, the use of HBO allowed for faster resolution of the infection. In the case of gas gangrene, a negative bacteriological result (direct preparation) was obtained after an average of eight treatments (4 days). As an adjunctive treatment for treating infections with mixed aerobic and anaerobic flora of soft tissues, such as gas gangrene, HBO was found to allow faster control of a rapidly developing infection. Assisted by surgical excision of necrotic fragments of fascia, skin, and subcutaneous tissue, drainage, and guided antibiotic therapy, it facilitates wound debridement to the point where defects can be repaired with grafts or skin flaps. It also contributes in many cases to reducing the extent of amputation in the affected limb [23]. Another review of experiences with hyperbaric oxygen (HBO) treatment of pediatric patients was published by the Israel Naval Medical Institute. In 1980 and 1997, 139 pediatric patients between the ages of two months and 18 years (mean age: 7.7 years) received HBO treatment at the institute. In this group of patients, 13 children (9.2%) were treated after crush, post-traumatic ischemia or impingement syndrome, four (2.8%) had myonecrosis caused by Clostridium, and one (0.7%) had necrotizing fasciitis. The outcome was assessed based on neurological complications, mortality, and degree of soft tissue loss and limb amputations. The positive effects of using HBO therapy were confirmed for 129 of the 139 pediatric patients included in the study [28]. 

### 3.4. Osteonecrosis

The efficacy of HBO therapy was also confirmed in a retrospective study conducted between 1988 and 2001 at the Department of Pediatric Oncology, Hematology, and Immunology at the Heinrich Heine University Düsseldorf on the treatment of osteonecrosis (ON) in pediatric cancer patients. Nineteen of 27 patients with acute lymphoblastic leukemia (ALL) between the ages of 7 months and 16 years were treated with HBO, with an average of 45 treatments. In children younger than 10 years, a decrease in the incidence of ON and conversion of aseptic osteonecrosis to bone edema has been reported [25]. Analogous results were obtained in a retrospective study published in 2000 at the Institute of Diagnostic Radiology at the Heinrich-Heine University, which analyzed 72 MRIs of 20 children undergoing chemotherapy with aseptic osteonecrosis. Twelve of the 20 patients were treated with HBO. Each session lasted about 130 min for each of the 12 patients. An initial compression phase lasting 10 min was followed by an inhalation phase with 100% oxygen at 2.5 ATA lasting 90 min. The session ended with a 10-min decompression phase. Oxygen inhalation was discontinued after 30 min for a 10-min phase of atmospheric air inhalation. The duration of HBOT treatment ranged from 6 to 12 weeks. It was noted that in the advanced stage of chemotherapy related to ON, HBO therapy had a significant role in pain relief [26]. Another study retrospectively evaluated 495 children and adolescents who were ALL patients diagnosed and treated at the Department of Pediatric Hematology and Oncology at Padua University Hospital from September 2000 to February 2017. Twenty-three of the 495 patients were diagnosed with ON. It was noted that the incidence of ON was associated with older age and higher doses of steroids. All patients underwent standard treatment, whereas eight of the 23 patients received HBOT. HBO therapy consisted of daily sessions of three times of 25 min each with 100% oxygen at 2.5 ATA alternating with atmospheric air for 5 min, with at least 30 consecutive sessions [21]. The patients were treated with HBO therapy, which proved safe and effective for most patients, even those who were immunocompromised or critically ill. Initial experiences with HBO (three patients included in this study and treated until 2014) provided the rationale for including HBO therapy in the treatment process. Joint replacement, recommended for all included children, was not applied in a third of them, who were nevertheless able to function almost normally. This means that, at least in some patients, the pain and functional limitations were tolerable, which was also demonstrated during clinical follow-up [21]. The study found that only children in the early stages of ON experienced the positive effects of HBO treatment. Three of the four patients with mild ON (Association Research Circulation Osseus—ARCO 1,2) treated with HBO therapy did not progress either clinically or on MRI. Half of the children were identified in the late stages (ARCO 3,4) and most had advanced symptoms (Common Terminology Criteria for Adverse Events—CTCAE 3). Four of them were treated with HBO therapy, but without success [21].

### 3.5. Thermal Burns

The effects of treating burns with HBO were reported in 1978 by Grossman, who conducted a study on a group of 381 patients. The age of the patients included in the study ranged from 9 months to 80 years. The HBO therapy used consisted of administering pure oxygen at a pressure of 2.5 to 3 ATA as follows: burns with an area greater than 50% total body surface (TBS): 90 min at 2.5 ATA three times a day for 7 days; burns with an area of 20% to 50% TBS: 90 min at 2.5 ATA twice a day for 7 days; and children: 45 min at 2 ATA twice a day for 7 days [29]. Grossman demonstrated that HBO therapy reduced the length of hospitalization compared to statistics published by the American Burn Association by up to ca. 55% for burns of less than 30% of the total body surface and resulted in lower complication and mortality rates and a reduction in the need for fluids by at least 30% [29]. The benefits of using hyperbaric oxygen therapy to treat burns were also confirmed by the Burn Treatment Center in Siemianowice Ślaskie in 2002–2006. Treatment in the chamber included 78 patients with thermal burns aged 10 months to 86 years. An analysis of the clinical material based on the diagnoses was performed, and the results obtained were analyzed and compared with a group of patients treated during the same period, for the same reason, and without HBO [23]. From 14 June 2002 to 26 May 2006, hyperbaric sessions were held in monoplace chambers at ETC Bara-Med. Pressure ranged from 2 to 2.5 ATA, with session durations ranging from 55 to 122 min. In the period from 29 May 2006 to 27 July 2006, hyperbaric sessions were conducted in a multiplace chamber and periodically in monoplace chambers. ETC Bara-Med monoplace chambers and a Haux Starmed 2500 multiplace chamber (two compartments of eight places each) were used at 2.5 ATA, with session durations ranging from 70 to 90 min [23]. In the group of burn patients treated without HBO, the average length of hospitalization was 5 days longer compared to that of HBO-treated patients. The mean time to prepare burn wounds for skin grafting was 8 days shorter compared to the group in which hyperbaric oxygen was not used. In a group of patients with burns to the face, head, and neck, hyperbaric oxygen therapy reduced the duration of swelling by an average of 23 h [23].

### 3.6. Skin Grafts

A retrospective study of pediatric patients who presented with facial injuries between 2008 and 2018 was conducted to analyze the effect of HBOT on tissue grafts. Tissue grafts were performed on the children, and the patients then received HBO therapy. In the analysis, a large composite graft was defined as a chondrocutaneous skin graft equal to or larger than 1.5 cm in diameter. Eight children with ear and nose injuries and one child with a congenital nasal deformity were identified. The need for HBOT was determined in patients early following the surgery. The HBO sessions began 24 h after the surgery. They were performed twice a day and each session lasted 90 min. The pressure in the chamber was 2.4 ATA. Patients remained in the facilities for the duration of HBO treatment and because of the ongoing need to monitor the graft. The number of sessions was determined by the clinical condition and was at most 20. Additional measures included cooling the graft with local compresses for 15–20 min every 1 to 2 h. The study confirmed the positive effect of HBO as an adjunctive therapy after grafting [19]. Another study looked at the use of HBOT in children after tissue transplantation associated with hypospadias. One study evaluated the post-transplant status of boys with hypospadias undergoing staged tubularized autograft repair (STAG). Patients receiving HBO pretreatment were compared with patients receiving a standard surgical procedure without HBOT. The HBOT protocol consisted of one session a day, five days a week for four weeks prior to surgery and 10 additional sessions immediately after the first stage of surgery. Each HBOT session consisted of 90 min of exposure to 100% oxygen at 2 ATA, with five-minute breaks every 20 min when atmospheric air was administered instead of oxygen. Seven boys received HBOT and 14 boys were in the control group. All patients in the HBOT group reported graft acceptance. In the control group, 57% of the patients had surgery without complications and were able to undergo the second stage of surgery, while 43% had graft shrinkage. With the exception of one patient (who was claustrophobic when entering the chamber), there were no significant side effects during HBOT. The study concluded that HBOT can be used safely to treat hypospadias in pediatric populations and reduce postoperative complications [18]. Another study tested whether a combination of NTG and HBOT would provide better outcomes after hypospadias surgery and would minimize complications and provide better results compared to NTG alone. In 2014–2019, 82 patients (2–24 years old) exhibiting varying degrees of scarring of the skin and subcutaneous tissue underwent reoperation to address complications of hypospadias after failed surgeries (three to nine surgeries, with an average of 5.5 being unsuccessful). Patients were divided into two groups: group I, consisting of 49 patients, received trimodal therapy that included NTG, HBOT, and locally administered steroids. Patients were examined every 3 weeks and then every 3 months. The surgical site was photographed by the parents or an older patient before each appointment. Group II, consisting of 33 patients, received NTG and steroids but did not receive HBO therapy. In group I: 44/49 (88.8%) of the surgeries were successful. Complications in this group included distal lesions (two cases) and urethral fistula (three cases). In Group II, successful outcomes were recorded in 23/33 cases (69.6%). Patients were observed from 5 months to 4 years after surgery. The study confirmed that treatment by combining HBO and NTG therapy results in improved tissue oxygenation and better wound healing [24].

## 4. Discussion

Hyperbaric oxygen therapy is one of the earliest developed medical technologies that has been in continuous use until today. Its history dates back nearly 350 years, making it a relatively well-researched method that is popular for treating many conditions [30]. At the Tenth European Consensus Conference on Hyperbaric Medicine in Lille on 15–16 April 2016, the use of HBOT was qualified as a recommendation for the treatment of specific conditions in pediatric patients: CO poisoning—Grade I recommendation, Level of Evidence B; necrotizing soft tissue infections—Grade I recommendation, Level of Evidence C; osteonecrosis—Grade II recommendation, Level of Evidence B; thermal burns—Grade II recommendation, Level of Evidence C; skin and skin flap grafts—Grade II recommendation, Level of Evidence C [16]. HBOT has also recently been reported to have a positive effect on adult patients with COVID-19. They experience an apparent increase in alveolar pressure, resulting in an increase in the rate of oxygen diffusion compared to standard methods [31]. Unfortunately, no direct references to the use of HBOT in the treatment of pediatric patients with COVID-19 and its complications have been found in the literature. However, the authors of the present study believe that studies conducted on an older population can be the basis for the analysis of the positive effects of HBOT on pediatric patients following COVID-19 who are struggling with complications. Knowing the general effects of hyperbaric oxygen therapy including the alleviation of general inflammation by reducing the expression of TLR2 and TLR4 receptors and polarizing cytokine-secreting macrophages (and thus preventing acute respiratory failure), reducing fatigue and improving cognitive abilities, reducing tissue edema and improving tissue oxygenation, and antimicrobial activity [14,32,33], it can be expected that its effect will also be positive in the treatment of children with the pediatric inflammatory multisystem syndrome (PIMS, MIS-C) developed as a complication of COVID-19, and the treatment of other complications such as pediatric acute respiratory distress syndrome (PARDS). Both the occurrence of MIS-C and PARDS are likely to be initiated by the secretion of large amounts of pro-inflammatory cytokines and lead to multi-organ dysfunction [9]. Therefore, it can be speculated that hyperbaric oxygen therapy could potentially be used to treat pediatric patients with COVID-19 to prevent the development of pediatric acute respiratory failure and pediatric inflammatory multisystem syndrome. All studies discussed in this paper showed reproducible and beneficial therapeutic effects of HBOT in the treatment of various conditions of pediatric patients, which leads to the conclusion that it is reasonable to use HBOT in the treatment of pediatric patients. However, to better understand the efficacy of the use of hyperbaric oxygen in the treatment of other conditions and to discover new opportunities offered by this form of treatment, more research is needed, especially on the impact of the therapy on the treatment of pediatric patients with COVID-19 and on those struggling with its complications. 

## 5. Conclusions

No evidence supported by research has been found in scientific journals on the effectiveness of the use of hyperbaric oxygen therapy in children with a history of COVID-19 infection.

The bibliographic review showed that hyperbaric oxygen therapy can be used in the treatment of children after carbon monoxide poisoning, with soft tissue necrosis, bone necrosis, after burns, and after skin transplants.

## Figures and Tables

**Figure 1 ijerph-19-15213-f001:**
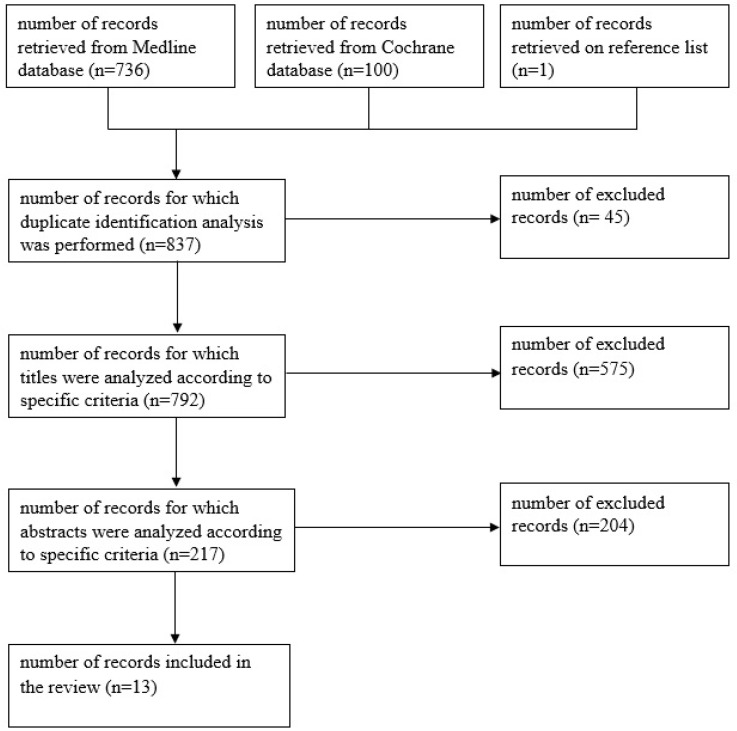
Figure describing the course of successive stages of including publications in the analysis.

**Table 1 ijerph-19-15213-t001:** A detailed summary of study discussed.

No.	Study	Type of Study	Disease Entity	Population	Intervention	Comparative Intervention	Measurement of the Final Result	Result for HBOT	Ref.
1	Yildiz et al. (2021)	retrosp.	CO poisoning	*n* = 93 TG *n* = 238 CG age < 19 years	HBOT	NBO treatment	Glasgow score, leukocyte count, troponin T level, carboxyhemoglobin level	- significantly better indices after HBO therapy	[17]
2	Neheman et al. (2020)	retrosp.	skin grafts	*n* = 7 TG *n* = 14 CG age < 18 years	- one HBOT session per day for 5 days during the 4 weeks before surgery and 10 additional sessions following the surgery - session duration: 90 min exposure to 100% oxygen at a pressure of 2 ATA with 5 min air breathing breaks every 20 min	standard treatment	successful when graft acceptance occurred and graft shrinkage was not observed	- positive result: all patients in the study group had good graft acceptance without shrinkage; in the control group, 57% had good graft acceptance and were able to proceed to the second stage, while 43% developed graft shrinkage	[18]
3	Camison et al. (2020)	retrosp.	skin grafts	*n* = 9 TG age: 1.6–15.1 years (mean age: 8.4 years)	- 2 HBOT sessions per day, maximum 20 sessions - session duration: 90 min at 2.4 ATA	not applicable (no control group)	successful when graft acceptance occurred without complications	- a positive result was observed in all patients	[19]
4	Chang et al. (2020)	retrosp.	skin grafts	*n* = 49 TG *n* = 33 CG age: 2–24 years	HBOT	standard treatment (vasodilators and topical steroids)	- examination of patients every 3 weeks, then every 3 months - successful when graft acceptance occurred without complications	- in the study group, 44/49 grafts were successful - in the control group, 23/33 grafts were successful	[20]
5	Biddeci et al. (2019)	retrosp.	osteonecrosis	*n* = 8 TG *n* = 15 CG age < 18 years	- at least 30 HBO treatments at a pressure of 2.5 ATA - a course of sessions: 3 periods of 25 min with 100% oxygen alternated with 5-min periods in which air was administered	standard treatment	- MRI evaluation	- at an advanced stage of ON: - no effect of HBO therapy in MRI in the early stage of ON: - pain relief, - disability relief, - positive effect on changes in MRI morphology	[21]
6	Teksam et al. (2019)	retrosp.	CO poisoning	*n* = 16 TG *n* = 29 CG age < 19 years (mean age: 9 years)	- HBOT session at 5 ATA, session duration: 90 min	NBO treatment	assessment of oxidative stress levels and antioxidant parameters based on the examination of blood and urine samples; samples collected: - at admission, - after normobaric oxygen treatment - after hyperbaric oxygen treatment	- no significant effect of HBO therapy on oxidative stress and antioxidant parameters was observed - no significant effect of NBO therapy on oxidative stress and antioxidant parameters was observed	[22]
7	Kawecki et al. (2016)	retrosp.	e.g., necrotizing soft-tissue infection, thermal burns	*n* = 43 TG *n* = 78 TG age: 10 months—86 years	- HBOT session at 2 to 2.5 ATA, - session duration: 55 to 122 min (monoplace chambers) and 70 to 90 min (multiplace chambers)	standard treatment	evaluation of the mean time of hospitalization, average time of skin preparation for graft; time of swelling	- reduced length of hospital stay - reduced time to prepare skin for the graft - reduced time of swelling	[23]
8	Yarar et al. (2008)	retrosp.	CO poisoning	*n* = 38 TG *n* = 36 CG age: 1–17.8 years	- HBOT session at 2.4 ATA - session duration: 140 min, including 100 min under pressure	NBO treatment	assessment of carboxyhemoglobin levels and period of hospitalization	- significantly lower carboxyhemoglobin levels - significantly shorter hospitalization period	[24]
9	Bernbeck et al. (2004)	retrosp.	osteonecrosis	*n* = 19 TG *n* = 8 CG mean age: 8.2 ±4.7 years (range: 7 months to 16 years)	- an average of 45 HBO treatments per patient (min. 13, max. 80 treatments)	standard treatment	- MRI scan evaluation performed every 3 months for the radiological extent of lesions - assessment of pain severity and location	children < 10 years old—beneficial effect of HBOT: - decrease in AON - conversion of AON to BME children > 10 years old—no beneficial effect on BME	[25]
10	Scherer et al. (2000)	retrosp.	osteonecrosis	*n* = 12 TG *n* = 8 CG mean age TG 10.9 ± 4.4 years mean age CG 12.3 ± 5.5 years	- an average of 43 HBO treatments at 2.5 ATA over a period of 6–12 weeks - session duration: 130 min, including a 90-min oxygen inhalation period	standard treatment	- MRI scan evaluation on a scoring system (1–6 points) independently by two radiologists - evaluation criteria included size, number, and location of AON	at an advanced stage of AON: - the positive role of HBOT in pain relief - no significant changes in MRI morphology in the early stage of AON: - positive effects on discrete forms of AON, especially on bone edema	[26]
11	Chou et al. (2000)	retrosp.	CO poisoning	*n* = 150 TG age < 18 years mean age: 7.2 years	HBOT	NBO treatment	assessment of carboxyhemoglobin levels and death rates	- a significantly lower death rate for children with symptoms of CO poisoning, without soot in the respiratory tract - significantly lower carboxyhemoglobin levels	[27]
12	Weisman et al. (1998)	retrosp.	(among others) CO poisoning, thermal burns, necrotizing soft-tissue infections	*n* = 139 TG age: 2 months—18 years (mean age: 7.7 years)	HBOT	standard treatment	assessment based on neurological complications, mortality, and degree of soft tissue loss and limb amputations	- positive results in 129 of 139 patients undergoing HBO	[28]
13	Grossman (1978)	retrosp.	thermal burns	*n* = 138 TG *n* = 243 CG age: 9 months—80 years	- 2 HBO treatment sessions per day at 2 ATA for 7 days, then once a day - session duration: 2 times a day for 45 min; once a day for 90 min	standard treatment	- children were examined once a day by a pediatrician - evaluation criteria included fluid requirements, duration of hospitalization, complication rate, mortality rate	- fluid requirements reduced by at least 30% - shorter hospital stay relative to the statistical length of hospital stay: by about 55% for burns below 30%, by about 23% for burns from 30 to 60%, by about 12% for burns above 60% - the complication rate decreased dramatically - the mortality rate decreased	[29]

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
