# Peer review of "Can Hyperbaric Oxygen Therapy Be Used to Treat Children after COVID-19? A Bibliographic Review"

_ijerph, 2022, doi:10.3390/ijerph192215213_

Round 1
Reviewer 1 Report
It is a very interesting paper.
However the title may be inappropriate. Indeed you are looking to find out if there is anything written on covid in children and HBO. Finding no article on this subject you describe other pathology supported by HBO, and you conclude by saying that "No evidence supported by research has been found in scientific journals on the effectiveness of the use of hyperbaric oxygen therapy in children with a history of Covid-19 infection."
The title could be:
"Can Hyperbaric Oxygen Therapy Be Used to Treat Children? and after COVID-19 Infection? A Bibliographic Review"
And the conclusion could be "The bibliographic review showed that hyperbaric oxygen therapy can be used safely in the treatment of children after carbon monoxide poisoning, with soft tissue necrosis, bone necrosis, after burns and after skin transplants.
No scientific article on the effectiveness of HBO on Covid-19 infection has been found in the literature"
Author Response
Dear Reviewer.
On behalf of the Team, we would like to thank you for your in-depth review of the article "Can hyperbaric oxygen therapy be used in the treatment of children after COVID-19 infection? Bibliographic review"
We thank the reviewers for their time reviewing our article and providing comments. It was your valuable and insightful comments that led to possible improvements in the current version.
The authors carefully considered the comments and tried to respond to each of them. We hope that the manuscript, after careful corrections, will meet your high standards.
The answers are given below point by point.
All modifications in the manuscript are highlighted in red.
Reviewer 1
General comment: This is a very interesting article.
Answer: Thank you very much
Comment 1 "... However, the title may be inappropriate. Indeed, you want to find out if there is anything written about covid in kids and HBO. Without finding an article on the subject, you describe another HBO-supported pathology and end by stating that "No evidence has been found to be supported by research in scientific journals on the effectiveness of hyperbaric oxygen therapy in children with a history of Covid-19 infection."
The title could be:
"Can hyperbaric oxygen therapy be used to treat children? and after COVID-19 infection? Bibliographic review "
Answer: Thank you for your kind suggestion. We have revised it accordingly.
Comment 2 "... And the conclusion could be:" A bibliographic review has shown that hyperbaric oxygen therapy can be safely used in the treatment of children after carbon monoxide poisoning, soft tissue necrosis, bone necrosis, burns and skin grafts.
No scientific article on the effectiveness of HBO in Covid-19 infection has been found in the literature "
Answer: Thank you for your kind suggestion. We have revised it accordingly.
Reviewer 2 Report
It is a very interesting article, but it needs revision regarding its methodology.
Line 3: "A Bibliographic Review" should change to "A Systematic Review".
Line 14: Instead of "this study used the systematic review method", you should mention if you followed the PRISMA Statement for systematic reviews.
Line 17: You must add the "Hyperbaric oxygen therapy (HBOT)" instead of only the abbreviation.
Lines 18-20: The whole sentence must be rephrased.
Lines 80-82: You must provide a risk of bias assessment of the studies. You mentiont that you didn't assess the quality of the included studies due to their heterogeneity. That decision has nothing to do with the PRISMA. All your included studies are retrospective. The quantitative analysis is not needed for a systematic review. You don't need to mention that you didn't conduct a meta-analysis.
In Figure 1 you mention number of excluded records from the titles and number of excluded studies from the abstract. That is the same stage according to the methodology of systematic reviews. There is another stage that you do not mention, which is the review - full-text assessment. In that stage you mention the reasons for not inclusion of the studies you excluded.
You have to read the PRISMA Statement, as well as the Cohrane Handbook for Systematic Reviews and revise your whole manuscript.
Round 2
Reviewer 2 Report
Dear authors,
Thank you for your responses.
You provide a very complete bibliographic review regarding the hyperbaric oxygen therapy in treating children after SARS-COV-2 infection.
One suggestion. The Title should be more proper if it mentioned either "[...] after COVID-19 (without the term infection) [...]" or "[...] after SARS-COV-2 infection [...]".
Author Response
Dear Reviewer,
On behalf of the Team, we would like to thank you for your in-depth review of the article "Can hyperbaric oxygen therapy be used in the treatment of children after COVID-19 infection? Bibliographic review"
We thank you for reviewing our article and providing valuable comments.
The authors carefully considered the comments and tried to respond to each of them. We hope that the manuscript, after careful corrections, will meet your high standards.
The answers are given below point by point.
All modifications in the manuscript are highlighted in red.
[General comment] You provide a very complete bibliographic review regarding the hyperbaric oxygen therapy in treating children after SARS-COV-2 infection.
Answer: Thank you very much for your comment.
[Comment 1]: One suggestion. The Title should be more proper if it mentioned either "[...] after COVID-19 (without the term infection) [...]" or "[...] after SARS-COV-2 infection [...]".
Answer: Thank you very much for your attention. We corrected the title accordingly.